# CALIBRATED SELF-VERIFICATION FOR MULTIMODAL LLMS VIA ADVANTAGE-DECOUPLED PREFERENCE OPTIMIZATION

## ABSTRACT

Multimodal large language models (LLMs) have achieved significant advances through serial test-time scaling, which involves generating longer reasoning traces at test-time, yet this approach encounters performance bottlenecks. Consequently, parallel test-time scaling becomes as an alternative approach, which generates multiple candidate solutions in parallel and selects the best one. However, existing methods either focus solely on training generators or verifiers, which limits performance improvements. We propose **ADPO**, Advantage-Decoupled Preference Optimization, an RL framework that trains a unified policy to generate answers and self-verify via preference reward and decoupled advantages. To enhance the model's verification ability, we introduce preference reward using discrete group-adaptive ranking for binary outcomes and margin-based pairwise comparisons for continuous signals, yielding more stable learning and better calibrated self-verification scores. To address the gradient interference problem in joint training of generation and verification tasks, we introduce decoupled optimization with separate advantages and cross-task loss masking, effectively improving both generation and verification capabilities. Ablation studies show **+0.03** average improvement in verification AUC/AP metrics. ADPO achieves superior performance on multimodal math reasoning, image grounding and mobile agent tasks, with improvements of **+2.8%/+1.4%** on MathVista/MMMU, **+1.9%** cIoU on ReasonSeg, **+1.7%/+1.0%** step success rate on AndroidControl/GUIOdyssey.

## 1 INTRODUCTION

The reliability and trustworthiness of multimodal LLMs ensure accurate and consistent outputs across text, images, and audio, which is crucial for stable real-world deployments. High trustworthiness builds user confidence, promoting widespread adoption and safe operation of multimodal technologies in deployment environments. Test-time scaling serves as an effective method to enhance reliability by allocating additional computational resources during inference to improve output quality and consistency.

DeepSeek-R1 (Guo et al., 2025) and OpenAI-o1 (OpenAI, 2024) demonstrate that reasoning models achieve improved performance through serial test-time scaling by increasing the number of thinking tokens during inference in mathematics and coding domains. However, when transferring to multimodal domains, recent work has found that reasoning provides only limited performance improvements on image clssification, video understanding and visual spatial understanding (Li et al., 2025; Liao et al., 2025).These observations highlight the limitations of token-level scaling alone and motivate the development of alternative principles for parallel test-time scaling that more effectively support robust and efficient multimodal reasoning.

Repeated sampling combined with best-of-$N$ selection is another approach to test-time scaling. Existing methods often exclusively improve either the model's generation or verification capabilities. Training a generator and then majority voting at test-time (Wang et al., 2022), as well as training a dedicated multimodal verifier and using a base model as the generator at test-time (Sun et al., 2025), both approaches show limited performance improvements(see table 2). We propose a RL-based framework to train a unified policy model that learns to both generate and self-verify.

Figure 1: **Overview of ADPO.** Our advantage-decoupled preference optimization jointly trains content generation and self-verification via separate advantage computation and mutual loss masking, enabling reliable test-time scaling through best-of-$N$ selection across multimodal tasks. The unified verifier provides reliable scoring that enables effective test-time scaling via best-of-N selection, significantly improving performance across multimodal tasks.

To train a unified policy for generation and verification, binary rewards are problematic: they are sparse and threshold-sensitive, amplify class-imbalance in self-sampled data (often leading to model predictions collapsing to entirely 0 or entirely 1 outputs), and discard magnitude cues about answer quality. We instead use preference reward that enforce within-group ranking between scores and true quality. For discrete tasks, we split samples by answer reward into positive/negative sets and compute group-adaptive thresholds $s^+$ and $s^-$, encouraging positives $s > s^-$ and negatives $s < s^+$. For continuous tasks, we apply margin-$\gamma$ pairwise comparisons to yield a smooth win-rate signal. This replaces brittle absolute thresholds with relative signals robust to imbalance, provides denser and more stable supervision, avoids hard-threshold information collapse, and improves score calibration and best-of-N selection—well suited for unified multimodal policy training.

As generation and verification are coupled, training suffers from two pitfalls: gradient interference—verification errors tug the generator (and vice versa), encouraging "reward hacking" that inflates scores instead of improving answers—and class imbalance in self-sampled data, where binary supervision drives the scoring head to 0/1 collapse, harming stability and calibration. We address this with dual-advantage optimization under a GRPO objective: compute separate advantages for generation and verification from content and preference reward, and apply mutual loss masking so each segment backpropagates only through its own tokens. Concretely, a single policy first outputs an answer and then a self-verification score; the generation segment uses verifiable answer rewards, while the verification segment uses preference reward positive/negative grouping with adaptive thresholds for discrete tasks, and margin-based pairwise comparisons for continuous tasks—to align scores with true quality. This decoupling suppresses reward hacking and gradient contamination, mitigates collapse under imbalance, stabilizes training, and yields better-calibrated, more discriminative scores, improving area under the ROC curve (AUC), average precision (AP) and best-of-N selection—delivering reliable test-time gains with only 10% extra training cost over a GRPO-only generator.

Our contributions are summarized as follows:

**1. Unified Preference reward.** We develop unified preference reward that maintain informativeness under severe class imbalance that improve calibration and are robust to class imbalance.

**2. Decoupled advantage optimization.** We introduce a principled approach to disentangle content generation and verification learning within a unified GRPO framework.

**3. Comprehensive validation.** Our method significantly improves task performance and verification quality: best-of-8 selection achieves +2.8/+1.4 accuracy gains on MathVista/MMMU, +1.9 cIoU on ReasonSeg, and +1.7/+1.0 step success rates on AndroidControl/GUIOdyssey.

## 2 RELATED WORK

**Reasoning and Test-Time Scaling.** Recent work scales reasoning at test time via longer thinking tokens and majority voting for LLMs (Guo et al., 2025; OpenAI, 2024; Wang et al., 2022; Shao et al., 2024). Multimodal variants adapt this paradigm with R1-style objectives and structured CoT

for VLMs (Liu et al., 2025b; Peng et al., 2025; Liu et al., 2025c; Shen et al., 2025; Huang et al., 2025a; Zhang et al., 2025a; Yang et al., 2025). In agentic settings, GUI agents adopt RL with explicit reasoning traces (Lu et al., 2025; Liu et al., 2025a; Qin et al., 2025; Huang et al., 2025b; Zhang et al., 2025c; Gu et al., 2025). However, recent "no-think" results suggest that more internal tokens do not always translate to better multimodal reasoning (Li et al., 2025; Liao et al., 2025). We instead couple solution generation with a learned self-verification signal, enabling reliable performance scaling through best-of-$N$ selection without fragile dependence on longer chains.

**Multimodal Reward Modeling and Generative Verifiers.** Another line studies reward modeling for multimodal alignment, including RLHF-style pipelines and chain-of-thought verification (Zhang et al., 2025b; Sun et al., 2025). Process or scalar reward models provide step-level or outcome supervision for reasoning (Du et al., 2025; Cao et al., 2025; Wang et al., 2025). Generative verifiers and LLM-as-judge train models to both solve and judge (Zhang et al., 2024; Zheng et al., 2023). In contrast, we use reinforcement learning to train a single policy for answer and calibrated confidence with separate advantages and mutual masking, and we do not finely control the positive/negative ratio in training data; instead, we employ preference reward rather than binary reward, enabling dependable best-of-$N$ across multimodal tasks.

## 3 METHOD

We propose *ADPO* (Advantage-Decoupled Preference Optimization) (see fig. 1), a framework that intergrates unified preference reward and advantage-decoupled optimization for reliable self-verification. Given a multi-modal query, our method first produces an answer and then outputs a self-verification score. At test time, we perform batch decoding to produce multiple candidate answers and select the answer with the highest self-verification score as the final output. This unified generation and verification paradigm achieves reliable self-verification without additional reward models.

### 3.1 PRELIMINARY

**GRPO.** For each question $q$, the behavior policy $\pi_{\theta_{\text{old}}}$ samples a *group* of $G$ responses $\{o_i\}_{i=1}^G$, where each response $o_i = (o_{i,1}, \ldots, o_{i,|o_i|})$ is a token sequence of length $|o_i|$ and assigned a sequence-level reward $R_i$. GRPO estimates advantages by *normalizing* rewards within each group and optimizes the current policy $\pi_\theta$ with a PPO-style clipped objective:

$$\mathcal{J}_{\text{GRPO}}(\theta) = \mathbb{E}\left[\frac{1}{G}\sum_{i=1}^{G}\frac{1}{|o_i|}\sum_{t=1}^{|o_i|}\left(\min\left(r_{i,t}(\theta)\hat{A}_{i,t}, \text{clip}(r_{i,t}(\theta), 1-\varepsilon, 1+\varepsilon)\hat{A}_{i,t}\right) - \beta D_{\text{KL}}(\pi_\theta \| \pi_{\text{ref}})\right)\right], \quad (1)$$

where $r_{i,t}(\theta) = \frac{\pi_\theta(o_{i,t}|q, o_{i,<t})}{\pi_{\theta_{\text{old}}}(o_{i,t}|q, o_{i,<t})}$ is the likelihood ratio, $\hat{A}_{i,t} = \frac{R_i - \text{mean}(\{R_i\}_{i=1}^G)}{\text{std}(\{R_i\}_{i=1}^G)}$ is the group-normalized advantage, $\varepsilon$ is the clipping parameter, $\beta$ is the KL coefficient, $D_{\text{KL}}$ is the KL regularization, and $\pi_{\text{ref}}$ is the reference policy.

### 3.2 UNIFIED PREFERENCE REWARD

As shown in fig. 2, we propose a *unified preference reward* framework that brings heterogeneous tasks—discrete and continuous—under a single supervision scheme. UPO couples (i) an answer-level reward that standardizes task feedback and (ii) a verifier-driven preference signal that improves discriminative ability and generalizes to continuous metrics.

**Answer Reward.** We unify answer rewards across heterogeneous tasks by factorizing them into correctness and quality components. For task $t$ with answer $y$ and ground truth $y^*$, we define the answer reward as:

$$R^a(y, y^*) = \mathcal{X}_t(y, y^*) \cdot \mathcal{G}_t(y, y^*), \quad (2)$$

where $\mathcal{X}_t \in \{0, 1\}$ denotes binary correctness and $\mathcal{G}_t \in [0, 1]$ represents quality credit. For discrete tasks such as mathematical reasoning and agent navigation, we set quality credit to unity and determine correctness through rule-based matching:

$$\mathcal{X}_{\text{math/agent}}(y, y^*) = \text{match}(y, y^*), \qquad \mathcal{G}_{\text{math/agent}}(y, y^*) = 1, \quad (3)$$

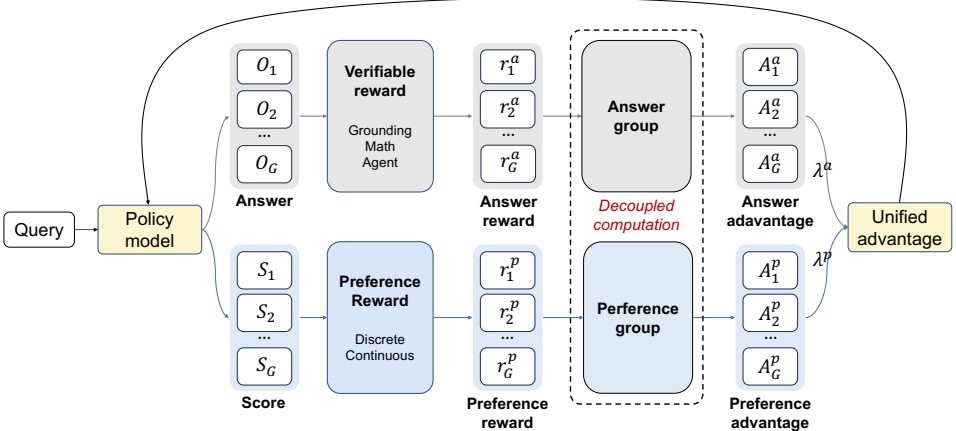

Figure 2: **The framework of ADPO.** Given an input, the policy model produces an answer and a self-verification score used to rank candidates. ADPO optimizes two complementary objectives: answer generation and self-verification. For answer generation, we introduce verifiable answer rewards that guide the model toward producing correct solutions. For self-verification, we design preference rewards that align the model's self-verification scores with ground-truth correctness. To prevent reward hacking and reduce coupling effects between the two processes, we employ separate advantage estimators and mutual loss masking under a GRPO objective.

Table 1: **Prompt for ADPO training.**

{Question} Output the thinking process in `<think></think>` and final answer choice(number) in `<answer></answer>` tags.
After outputting the answer, you will act as a correctness evaluation assistant and assign a score between 0 and 1 to indicate how accurate the answer is. If you believe the answer is correct, the score should be close to 1; otherwise, it should be close to 0.
For example:
`<think>`reasoning process here`</think>`
`<answer>`answer here`</answer>`
`<score>`score number here`</score>`.

where $\text{match}(\cdot, \cdot)$ implements task-specific equivalence checking. For continuous tasks like visual grounding, we treat all predictions as eligible and grade them by task-specific metrics:

$$\mathcal{X}_{\text{grounding}}(y, y^*) = 1, \qquad \mathcal{G}_{\text{grounding}}(y, y^*) = \text{IoU}(y, y^*), \tag{4}$$

where $\text{IoU}$ measures the spatial overlap between predicted and ground truth bounding boxes.

**Binary Reward.** To enable self-verification, we prompt the model to produce confidence estimate using dedicated instruction (see table 1). After generating the answer in `<answer></answer>`, the model outputs a confidence score $s \in [0, 1]$ in `<score></score>` indicating predicted correctness. We introduce a binary reward $R^b$ to calibrate model confidence against ground truth:

$$R^b(y, y^*) = \mathbb{1}\{(s > \tau) = \mathcal{X}_t(y, y^*)\}, \tag{5}$$

where $\tau$ is the binarization threshold. Thus, $R^b{=}1$ when the predicted score agrees with the ground-truth label and $R^b{=}0$ otherwise. Despite its simplicity, this consistency reward has three key limitations: (i) the binary score lacks sufficient discriminative capability for effective answer selection; (ii) the prevalent correct predictions (see fig. 3a) creates a pronounced class imbalance, incentivizing degeneration of always predicting $s{=}1$; (iii) it only applies to discrete tasks, as continuous tasks lack a well-defined binary correctness signal $\mathcal{X}_t$.

**Preference Reward.** To address these limitations, we introduce a preference-based reward that provides contrastive supervision and naturally extends to continuous tasks. Intuitively, a positive sample should be rewarded when its verification score exceeds the average of its negative counterparts, and vice versa. The core idea is to *adaptively partition* samples into positive and negative groups and *maximize their margin* to enhance discriminative capability. For the $i$-th sample with answer $y_i$,

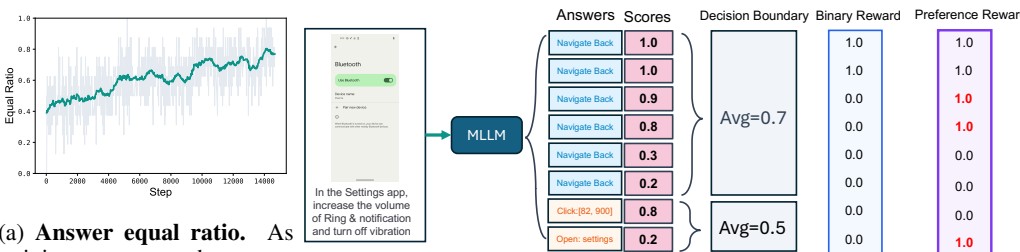

(a) **Answer equal ratio.** As training progresses, the answer reward all equal ratio within the group increases. Under binary rewards, this yields increasingly sparse gradients and pushes the model's self-verification scores toward a degenerate 0/1 output.

(b) **Discrete preference reward.** Within each group, we compute the average verification score of the positive set and the negative set and use these as data-driven boundaries. A positive sample receives a reward of 1 if its score is higher than the negatives' average; a negative sample receives a reward of 1 if its score is lower than the positives' average; otherwise the reward is 0. This preserves relative score magnitudes, avoids hard-threshold collapse, and improves calibration for best-of-$N$ selection.

ground truth $y_i^*$, and verification score $s_i$, we define the preference reward $R_i^p$ as:

$$R_i^p(y_i, y_i^*, s_i) = \mathbb{1}\{(s_i > \mu_i) = \mathcal{X}_t(y_i, y_i^*)\}, \tag{6}$$

$\mu_i$ is the mean verification score of the contrasting group:

$$\mu_i = \mathbb{E}_{j \in \{1,\dots,G\}}[\, s_j \mid \varphi_t(y_i, y_j)\,], \tag{7}$$

The function $\varphi_t(y_i, y_j)$ defines task-specific criteria for sample partitioning. For discrete tasks (*e.g.*, math reasoning, agent navigation), we partition samples based on answer correctness:

$$\varphi_{\text{math/agent}}(y_i, y_j) = \mathbb{1}\{\mathcal{X}_{\text{math/agent}}(y_i, y_i^*) \neq \mathcal{X}_{\text{math/agent}}(y_j, y_j^*)\}. \tag{8}$$

For continuous tasks (*e.g.*, visual grounding), we regard samples with similar quality as positves and others as contrastive negatives. We impose a margin $\gamma > 0$ on quality differences:

$$\varphi_{\text{grounding}}(y_i, y_j) = \mathbb{1}\{|\mathcal{G}_t(y_i, y_i^*) - \mathcal{G}_t(y_j, y_j^*)| > \gamma\}. \tag{9}$$

This preference reward provides dense, contrastive supervision that maximizes *quality-dependent score margins* while accommodating both discrete correctness and continuous paradigms.

**Unified Objective.** We optimize a unified objective that aggregates answer and preference rewards:

$$R_{\text{total}} = R^a(y, y^*) + R^p(y, y^*, s). \tag{10}$$

### 3.3 Advantage-Decoupled Optimization

During experimentation, we observed that joint optimization of generation and verification objectives creates conflicting gradients that degrade model performance. Answer rewards favor samples with higher response quality, while preference rewards favor samples with well-calibrated verification scores. To address this conflict, we decouple the advantage group with disjoint token masks. We compute separate advantages within each reward group: $\hat{A}^a$ from answer rewards and $\hat{A}^p$ from preference rewards. We then apply task-specific token masks to isolate gradients: $M^a$ covers answer generation tokens (including reasoning when present), while $M^p$ covers only verification score tokens. This prevents gradient interference between generation and verification objectives. The unified training objective becomes:

$$\mathcal{J}(\theta) = M^a \cdot \mathcal{J}_{\text{GRPO}}(\hat{A}^{(a)}) + M^p \cdot \mathcal{L}_{\text{GRPO}}(\hat{A}_{i,t}^{(p)}). \tag{11}$$

## 4 Experiments

We evaluate ADPO across three diverse multimodal domains: mathematical reasoning, visual grounding, and GUI agent tasks. Our experiments demonstrate that the ADPO consistently improves both task performance and self-verification reliability compared to existing methods.

## 4.1 Experimental Setup

**Datasets.** We evaluate on representative benchmarks across three domains: (1) **Multimodal math reasoning**: We train on multimodal-open-r1-8k-verified dataset (LMMs-Lab, 2025) and evaluate on MathVista (Lu et al., 2023) for in-domain performance and MMMU (Yue et al., 2024) for OOD generalization, focusing on the model's ability to perform math reasoning in visually grounded contexts with accuracy as the evaluation metric. (2) **Visual grounding**: We train on RefCOCO (Yu et al., 2016) and evaluate on ReasonSeg (Lai et al., 2024), focusing on referring expression comprehension with cIoU metrics. (3) **Mobile agent**: We train separately on AndroidControl (Li et al., 2024) and GUIOdyssey (Lu et al., 2024) training sets and evaluate on their respective test sets for mobile interface navigation, assessing step success rates(SR).

**Baselines.** We benchmark our method against three primary baselines: GRPO, GRPO with majority voting, and GRPO with LLM-as-judge verification. For the verification-based baseline, we evaluate three distinct LLM judges: the base model, the GRPO-trained model, and our ADPO-trained model. To specifically assess performance on mathematical tasks, we introduce an additional baseline using a reward model finetuned on specialized mathematical data.

**Implementation details.** All models are trained with a consistent set of hyperparameters: a learning rate of $1 \times 10^{-6}$, a batch size of 128, a group size $G = 8$, a GRPO clipping parameter $\varepsilon = 0.2$, and a KL divergence coefficient $\beta = 0.01$. For the **Multimodal Math Reasoning** task, we fine-tune Qwen2-VL-7B (Wang et al., 2024) for 1200 steps. For **Visual Grounding** and **Mobile Agent**, we use Qwen2.5-VL-7B (Bai et al., 2025) as the base model, training for 1200 and 8000 steps, respectively. During training rollout, we decode with temperature $T = 1.0$ and top-$p = 0.99$; at evaluation, we use $T = 0.2$ and top-$p = 0.99$.

## 4.2 Main Results

We evaluate three generators: the base model, the GRPO-finetuned model, and the ADPO-finetuned model, each paired with four verification strategies: majority voting, Qwen-as-judge, GRPO-as-judge, and ADPO-as-judge. We report pass@1 and best-of-N for $N \in \{4, 8, 12\}$. Across all domains, using ADPO as a unified generator and verifier yields the best performance under the same sampling budget, while preserving pass@1 generation quality comparable to GRPO.

ADPO enables effective self-verification with superior generation quality. Under equal sampling budgets (N=8 and N=12), ADPO delivers the strongest best-of-N on all three domains. At N=8, its improvement over the next best approach are at least +1.0 on MathVista, +0.1 on ReasonSeg, and +1.3 on AndroidControl; at N=12, the gains are at least +0.7, +0.3, and +1.4, respectively. These results show that ADPO adds robust verification while preserving single-sample quality.

ADPO delivers stronger best-of-$N$ performance. When used as both generator and verifier, it consistently surpasses GRPO and majority/LLM-as-judge baselines across sample budgets. On MathVista, ADPO climbs from 64.8 at $N=4$ to 65.3 at $N=12$, exceeding GRPO by 1.4-2.1 points and Base by 4.6-6.6 points across $N$. On ReasonSeg (overall cIoU), it improves from 61.1 to 61.6, maintaining 0.1-0.3 point gains over GRPO and 3.3-3.8 over Base. On AndroidControl (success rate), ADPO stays around 72.7-72.9, leading GRPO by 1.3-1.5 and Base by 8.0-12.0 points. These consistent margins over $N \in \{4, 8, 12\}$ show that ADPO yields more sample-efficient selection and higher best-of-$N$ returns than competing generator-verifier pairings.

ADPO equips the model with robust, cross-generator verification. Our judges remain strong even on outputs from weaker generators: on MMMU with $N = 8$ from the baseline generator, the ADPO judge reaches 51.2%, outperforming all baselines by +1.8-+6.2 points. On ReasonSeg, ADPO's judge improves from 60.9% to 61.6% as $N$ increases from 4 to 12, exceeding the GRPO judge at every budget by +0.9-+2.0 points. For GUI agents, ADPO judges lead on both AndroidControl and GUI Odyssey at $N \in \{4, 12\}$, with consistent gains of roughly +0.4-+0.7 points over GRPO. Calibration metrics align with these trends: ADPO yields higher AUC/AP across domains—for example, +1.8/+3.5 points on Math, +0.031/+0.030 on ReasonSeg, and +0.185/+0.126 on GUI—confirming superior verification quality.

Performance scales with the sampling budget $N$. With ADPO finetuned model as the generator, best-of-$N$ improves monotonically, and pairing it with ADPO finetuned model as the verifier matches or exceeds the strongest alternatives. On MMMU, performance increases $50.8 \rightarrow 52.1 \rightarrow 52.3$,

Table 2: **Evaluation results on multi-modal math reasoning benchmarks.** Rows correspond to generators and columns correspond to verifiers. We use Qwen2-VL-7B as the base model, with GRPO and ADPO representing the finetuned models. Majority voting serves as the verifier baseline. Models are trained on multimodal-open-r1-8k-verified (LMMs-Lab, 2025) dataset and tested on MathVista (Lu et al., 2023) for in-domain performance and MMMU (Yue et al., 2024) for OOD generalization. MM-Verify is a fine-tuned reward model used to select the best answer from multiple samples generated by the Qwen base model. See Appendix table 6 for detailed results.

| Verifier / Generator | MathVista (In-domain) | | | | MMMU (OOD) | | | |
| --- | --- | --- | --- | --- | --- | --- | --- | --- |
| | Major | Base | GRPO | ADPO | Major | Base | GRPO | ADPO |
| *Sample 1* | | | | | | | | |
| MM-RLHF | 61.6 | | | | - | | | |
| R1-VL-7B | **63.5** | | | | - | | | |
| Base | 57.9 | | | | 47.1 | | | |
| GRPO | 62.2 | | | | **48.7** | | | |
| ADPO | **62.4** | | | | 47.7 | | | |
| *Sample 4* | | | | | | | | |
| MM-Verify | **59.8** | | | | - | | | |
| Base | 58.2 | 55.7 | 55.5 | 56.4 | 48.6 | 45.2 | 45.8 | 49.9 |
| GRPO | 63.4 | 62.4 | 62.1 | 62.0 | 49.4 | 49.3 | 49.9 | 50.1 |
| ADPO | 63.3 | 61.5 | 62.1 | **64.8** | 50.7 | 48.3 | 49.1 | **50.8** |
| *Sample 8* | | | | | | | | |
| MM-Verify | **62.5** | | | | - | | | |
| Base | 60.1 | 57.0 | 56.4 | 56.5 | 49.4 | 45.0 | 46.6 | 51.2 |
| GRPO | 62.9 | 60.7 | 60.8 | 60.5 | 51.1 | 49.7 | 49.3 | 49.8 |
| ADPO | 64.0 | 62.3 | 62.3 | **65.0** | 51.8 | 51.6 | 51.2 | **52.1** |
| *Sample 12* | | | | | | | | |
| MM-Verify | **64.1** | | | | - | | | |
| Base | 60.7 | 56.9 | 56.3 | 55.0 | 50.7 | 45.8 | 45.2 | 50.6 |
| GRPO | 63.4 | 62.5 | 62.5 | 61.8 | 51.7 | 51.2 | 50.8 | 51.3 |
| ADPO | 64.6 | 63.0 | 63.5 | **65.3** | 51.2 | 52.0 | **52.6** | 52.3 |

Table 3: **Evaluation results on image grounding benchmarks.** Rows correspond to generators and columns correspond to verifiers. We use Qwen2-VL-7B as the base model, with GRPO and ADPO representing the finetuned models. Majority voting serves as the verifier baseline. Models are trained on RefCOCO (Yu et al., 2016) and tested on ReasonSeg (Lai et al., 2024) (out-of-domain). We report cIoU (%) on ReasonSeg for all methods. See Appendix table 7 for detailed results.

| Verifier / Generator | Short query | | | | Long query | | | | Overall | | | |
| --- | --- | --- | --- | --- | --- | --- | --- | --- | --- | --- | --- | --- |
| | Major | Base | GRPO | ADPO | Major | Base | GRPO | ADPO | Major | Base | GRPO | ADPO |
| *Sample 1* | | | | | | | | | | | | |
| LISA-7B | **48.5** | | | | 48.9 | | | | 48.8 | | | |
| SegLLM | - | | | | **54.2** | | | | 48.4 | | | |
| Seg-Zero-7B | - | | | | - | | | | **52.0** | | | |
| Base | 51.8 | | | | 57.0 | | | | 56.7 | | | |
| GRPO | 55.5 | | | | 59.7 | | | | 59.5 | | | |
| ADPO | **55.7** | | | | **60.2** | | | | **59.9** | | | |
| *Sample 4* | | | | | | | | | | | | |
| Base | 52.0 | 51.4 | 53.2 | 53.5 | 57.9 | 57.5 | 57.9 | 57.9 | 57.2 | 57.1 | 57.7 | 57.7 |
| GRPO | 57.0 | 57.5 | 56.4 | **57.8** | 59.5 | 60.3 | 59.7 | 61.1 | 59.4 | 60.2 | 59.5 | 60.9 |
| ADPO | 51.4 | 54.1 | 54.3 | 55.1 | 59.5 | 59.9 | 60.7 | **61.5** | 59.0 | 59.6 | 60.3 | **61.1** |
| *Sample 8* | | | | | | | | | | | | |
| Base | 51.4 | 51.4 | 50.5 | 50.9 | 57.8 | 57.2 | 57.5 | 58.4 | 57.4 | 56.9 | 57.0 | 57.9 |
| GRPO | 55.6 | 55.0 | 55.3 | 57.2 | 59.9 | 60.7 | 60.7 | 61.3 | 59.6 | 60.4 | 60.4 | 61.1 |
| ADPO | 55.2 | 55.4 | **57.8** | 55.9 | 58.2 | 60.2 | 60.6 | **61.5** | 58.0 | 59.9 | 60.5 | **61.2** |
| *Sample 12* | | | | | | | | | | | | |
| Base | 53.7 | 53.3 | 53.1 | 52.6 | 57.9 | 57.6 | 57.9 | 58.2 | 57.6 | 57.4 | 57.6 | 57.8 |
| GRPO | **58.1** | 52.4 | 55.8 | 56.7 | 59.5 | 60.2 | 59.9 | 61.6 | 59.4 | 59.7 | 59.6 | 61.3 |
| ADPO | 55.3 | 57.4 | 56.5 | 56.2 | 60.0 | 60.9 | 61.0 | **62.0** | 59.8 | 60.7 | 60.7 | **61.6** |

competitive with the GRPO judge (52.6 at $N = 12$). For ReasonSeg, the ADPO verifier achieves $61.1 \rightarrow 61.2 \rightarrow 61.6$, exceeding the next best (60.3/60.5/60.7). Overall, under the same generator, ADPO yields consistent improvements in best-of-$N$ as $N$ increases.

OOD results show ADPO's strong generalization. On MMMU, ADPO (gen+judge) reaches 52.1% at N=8, surpassing the best GRPO pairing (51.1%, majority) and the strongest baseline (51.2%, ADPO judge). Across N, it stays ahead: 50.8 at N=4 (+0.7-0.9%) and, with a GRPO judge, peaks at 52.6 at N=12. As a judge, ADPO also boosts weak Base generators (51.2% vs. 45.0-46.6%). On ReasonSeg, ADPO yields 61.1-61.6 cIoU (N=4/8/12), topping GRPO by +0.1-0.3% and baselines by

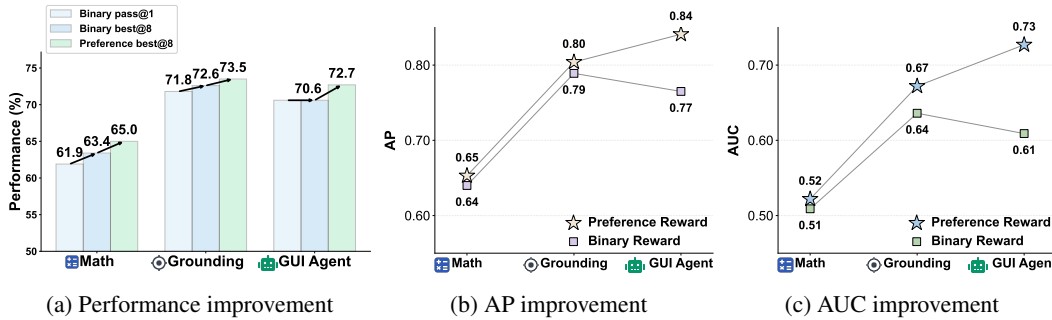

(a) Performance improvement  (b) AP improvement  (c) AUC improvement

Figure 4: **Ablation of Preference reward.** We compare Binary reward with Preference reward across three domains: Math, Grounding, and GUI Agent. (a) Preference reward achieves superior generation quality (relative to Binary pass@1) while consistently improving best@8. (b-c) Preference reward yields higher AP and AUC for self-verification, indicating better calibration of verification scores to correctness. Numbers above markers denote absolute values; error bars omitted for clarity. See Appendix table 9 for full results.

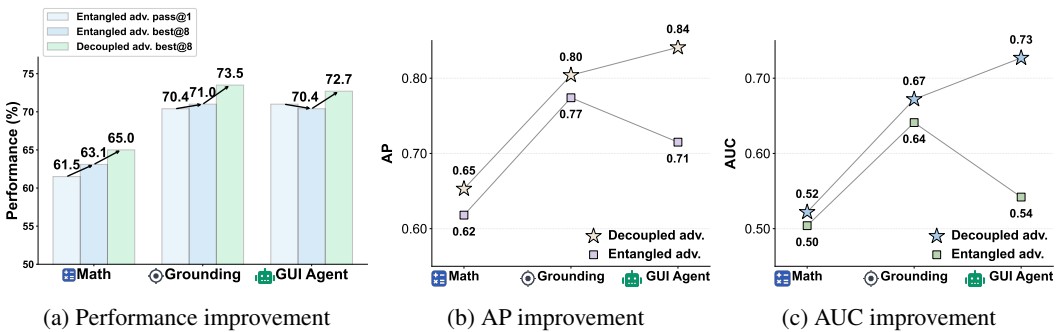

(a) Performance improvement  (b) AP improvement  (c) AUC improvement

Figure 5: **Ablation on decoupled advantages.** Across math, grounding, and mobile agent, decoupled advantages policy optimization achieves superior generation quality (pass@1) while yielding higher best@8 (a). It also improves the calibration of self-verification, reflected by higher AP (b) and AUC (c). See Appendix table 10 for full results.

+3.3-3.8%. Overall, unified training improves generator robustness and verifier calibration, enabling effective best-of-N selection under shift.

## 4.3 ABLATION STUDIES

We conduct comprehensive ablation studies to analyze the key components of our decoupled advantage preference optimization framework.

**Effect of preference reward.** Figure 4 shows the impact of preference reward compared to binary reward across all three domains. The preference formulation consistently improves both task performance and self-verification quality. For mathematical reasoning, we observe +1.6% improvement in best@8 performance and +1.3% improvement in average precision (AP). The benefits are even more pronounced for self-verification metrics, with AUC improvements of +1.3%, +3.6%, and +11.8% for math, grounding, and agent tasks respectively. This demonstrates that preference reward provide more stable training signals and better calibrated confidence scores, particularly important under the naturally imbalanced positive/negative distributions in self-verification learning.

**Effect of decoupled advantages.** Figure 5 illustrates the contribution of our decoupled advantage computation with mutual loss masking compared to simple reward aggregation. Decoupled advantages consistently outperform entangled advantages across all domains, with particularly significant improvements in self-verification quality. For GUI agent tasks, decoupled advantages achieve +2.8% improvement in best@8 performance and substantial gains in AUC +18.5 This validates our hypothesis that separating gradient flows between content generation and self-judgment prevents reward hacking and enables more effective optimization of both objectives.

Table 4: **Evaluation results on mobile agent benchmarks.** Rows correspond to generators and columns correspond to verifiers. We use Qwen2-VL-7B as the base model, with GRPO and ADPO representing the finetuned models. Majority voting serves as the verifier baseline. Models are separately trained on AndroidControl (Li et al., 2024) and GUIOdyssey (Lu et al., 2024) training sets and tested on respective test sets. We report step surpassing rate (%) on AndroidControl and GUIOdyssey for all methods. See Appendix table 8 for detailed results.

| Verifier / Generator | AndroidControl (SR) | | | | GUI Odyssey (SR) | | | |
|---|---|---|---|---|---|---|---|---|
| | Major | Base | GRPO | ADPO | Major | Base | GRPO | ADPO |
| *Sample 1* | | | | | | | | |
| UI-TARS-7B | | **72.5** | | | | 67.9 | | |
| SpiritSight-8B | | 68.1 | | | | **75.8** | | |
| AgentCPM-GUI-8B | | 69.2 | | | | 75.0 | | |
| Base | | 61.3 | | | | 52.8 | | |
| GRPO | | **71.0** | | | | **79.8** | | |
| ADPO | | 70.9 | | | | 79.7 | | |
| *Sample 4* | | | | | | | | |
| Base | 56.0 | 52.5 | 57.7 | 60.7 | 46.5 | 45.2 | 45.3 | 45.6 |
| GRPO | 71.0 | 71.0 | 70.8 | 71.2 | 81.3 | 81.0 | 80.7 | 81.4 |
| ADPO | 71.6 | 71.0 | 72.0 | **72.7** | 79.8 | 81.2 | 81.1 | **81.6** |
| *Sample 8* | | | | | | | | |
| Base | 58.3 | 54.3 | 61.0 | 64.7 | 46.6 | 44.9 | 44.5 | 44.6 |
| GRPO | 70.8 | 71.0 | 70.9 | 71.4 | 81.5 | 80.7 | 80.6 | 81.2 |
| ADPO | 71.3 | 70.8 | 71.4 | **72.7** | 80.9 | 81.6 | 81.4 | **81.7** |
| *Sample 12* | | | | | | | | |
| Base | 58.3 | 53.6 | 60.7 | 64.5 | 46.9 | 44.6 | 44.0 | 43.6 |
| GRPO | 71.1 | 71.4 | 70.9 | 71.5 | 81.1 | 79.9 | 79.7 | 80.3 |
| ADPO | 71.9 | 71.6 | 71.9 | **72.9** | 80.5 | **81.5** | 81.1 | 81.4 |

Table 5: **Ablation on the margin $\gamma$ for continuous preference rewards.** We evaluate different margin values on ReasonSeg. $\gamma = 0.100$ yields the bset overall gIoU(60.4) and cIoU(61.2).

| $\gamma$ | Short query | | | Long query | | | Overall | | |
|---|---|---|---|---|---|---|---|---|---|
| | gIoU | cIoU | ACC | gIoU | cIoU | ACC | gIoU | cIoU | ACC |
| 0.025 | **53.7** | 56.5 | **69.9** | 58.1 | 58.9 | 71.3 | 57.8 | 58.8 | 71.1 |
| 0.050 | 52.6 | 54.4 | 63.1 | 60.2 | 61.0 | 73.3 | 59.8 | 60.5 | 72.7 |
| **0.100** | 53.2 | 56.0 | 67.0 | **60.9** | **61.5** | **73.7** | **60.4** | **61.2** | **73.5** |
| 0.200 | 53.2 | 55.7 | 66.0 | 59.9 | 60.7 | 72.7 | 59.6 | 60.4 | 72.3 |
| 0.250 | **53.7** | **56.8** | 68.9 | 59.7 | 60.4 | 72.5 | 59.3 | 60.2 | 72.3 |

**Margin parameter analysis.** Table 5 analyzes the effect of margin parameter $\gamma$ in continuous preference reward computation for visual grounding tasks. We find that $\gamma = 0.1$ provides the optimal balance, achieving 73.5% overall accuracy. Too small margins ($\gamma = 0.025$) may not provide sufficient discrimination between similar quality outputs, while too large margins ($\gamma \geq 0.2$) may be overly restrictive and reduce the density of preference signals. This hyperparameter study confirms the importance of carefully tuning the preference threshold for optimal performance.

## 5 CONCLUSION

We introduce ADPO (Advantage-Decoupled Preference Optimization), a unified reinforcement learning framework that trains a single policy to both generate solutions and perform self-verification. ADPO addresses three key challenges in test-time scaling: (i) it enables reliable parallel best-of-N selection through unified generator-verifier training; (ii) it replaces binary supervision with relative, batch-adaptive preference reward that improve calibration across both discrete and continuous tasks; and (iii) it employs decoupled advantages to separate gradient flows for generation and verification, thereby mitigating reward hacking and gradient interference. Extensive evaluation across five benchmarks spanning three domains: MathVista, MMMU, ReasonSeg, AndroidControl, and GUI Odyssey, which demonstrates that ADPO achieves superior pass@1 performance while consistently improving best-of-N selection and delivering superior self-verification calibration.

ETHICS STATEMENT

This work adheres to the ICLR Code of Ethics. The study relies solely on publicly available datasets under their original licenses; no personally identifiable information is processed beyond what is already public, and all datasets were used in accordance with their terms. No new human-subject experiments were conducted; when third-party annotations are involved, they were collected by the dataset providers under their respective ethics approvals. We assessed potential risks of misuse (e.g., generating harmful or deceptive content) and release only evaluation scripts and models consistent with responsible-use guidelines. We report known failure cases and distributional limitations, and we caution against deploying our method in safety-critical settings without additional safeguards.

REPRODUCIBILITY STATEMENT

Due to company policy, the complete training code cannot be released at submission time. We will release the training code, inference and evaluation code upon approval.

**Models and hyperparameters.** All models are trained with a consistent set of hyperparameters: learning rate $1 \times 10^{-6}$, batch size 128, group size $G = 8$, GRPO clipping parameter $\varepsilon = 0.2$, and KL coefficient $\beta = 0.01$. For Multimodal Math Reasoning, we fine-tune Qwen2-VL-7B (Wang et al., 2024) for 1200 steps. For Visual Grounding and Mobile Agent, we finetune Qwen2.5-VL-7B (Bai et al., 2025) for 1200 and 8000 steps, respectively.

**Training data.** We use only publicly available training sets for each domain: multimodal-open-r1-8k-verified (LMMs-Lab, 2025) (math), RefCOCO (Yu et al., 2016) (grounding), AndroidControl (Li et al., 2024), and GUI Odyssey (Lu et al., 2024) (mobile agents). No proprietary or sensitive data are used.

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

# A  APPENDIX

## LLM USAGE DISCLOSURE

We used large language models (LLMs) to assist (i) software engineering tasks (code scaffolding, refactoring, and boilerplate generation) and (ii) academic writing (grammar polishing, tone adjustment, and minor phrasing improvements). All research ideas, algorithmic designs, experimental setups, and final claims are authored and verified by the human authors.

LLMs were not used to generate results, annotations, or proofs; they did not perform data labeling, hyperparameter selection, or evaluation. All code produced with LLM assistance was reviewed and unit-tested by the authors. For writing, we preserved factual accuracy, checked references, and rewrote any ambiguous outputs. We screened for potential plagiarism and license contamination and ensured that all cited text and code are properly attributed.

Table 6: **Evaluation results on multi-modal math reasoning benchmarks.** Rows correspond to generators and columns correspond to verifiers. We use Qwen2-VL-7B as the base model, with GRPO and ADPO representing the finetuned models. Majority voting serves as the verifier baseline. Values are accuracy (%). GVQA: General VQA; MVQA: Math Target VQA; ARD: Art & Design; BUS: Business; HEM: Health & Medicine; HSS: Human & Social Science; SCI: Science; TEN: Technology & Engineering.

| Generator | Verifier | MathVista (In-domain) | | | MMMU (OOD) | | | | | | |
|---|---|---|---|---|---|---|---|---|---|---|---|
| | | GVQA | MVQA | **ALL** | ARD | BUS | HEM | HSS | SCI | TEN | **ALL** |
| *Sample 1* | | | | | | | | | | | |
| Base | ✗ | 68.9 | 48.5 | 57.9 | **67.5** | 39.1 | 49.3 | 69.0 | 33.9 | 36.7 | 47.1 |
| GRPO | ✗ | **69.8** | 55.7 | 62.2 | 65.0 | 45.9 | 48.2 | 68.2 | **35.9** | **39.8** | **48.7** |
| ADPO | ✗ | 68.7 | **57.0** | **62.4** | 63.1 | **46.2** | **50.2** | **71.1** | 33.3 | 35.3 | 47.7 |
| *Sample 4* | | | | | | | | | | | |
| Base | Major | 65.7 | 51.9 | 58.2 | 66.7 | 47.3 | 50.7 | 65.8 | 34.0 | 38.1 | 48.6 |
| | Base | 63.9 | 48.7 | 55.7 | 60.0 | 44.0 | 50.7 | 60.8 | 32.7 | 33.8 | 45.2 |
| | GRPO | 63.3 | 48.9 | 55.5 | 61.7 | 43.3 | 50.0 | 63.3 | 30.0 | 36.7 | 45.8 |
| | ADPO | 63.3 | 50.6 | 56.4 | 66.7 | 49.3 | 53.3 | 70.8 | 34.0 | 37.6 | 49.9 |
| GRPO | Major | 69.8 | 58.0 | 63.4 | 65.8 | 44.7 | 50.0 | 70.0 | 42.0 | 36.7 | 49.4 |
| | Base | 70.2 | 55.7 | 62.4 | 62.5 | 44.0 | 50.0 | 70.8 | 40.0 | 39.5 | 49.3 |
| | GRPO | 69.6 | 55.7 | 62.1 | 62.5 | 44.0 | 50.0 | 70.8 | 40.0 | 39.5 | 49.9 |
| | ADPO | 69.6 | 55.6 | 62.0 | 64.2 | 46.7 | 50.7 | 71.7 | 39.3 | 39.5 | 50.1 |
| ADPO | Major | 71.7 | 56.1 | 63.3 | 66.7 | 48.0 | 52.7 | 70.0 | 39.3 | 39.0 | 50.7 |
| | Base | 68.5 | 55.6 | 61.5 | 64.2 | 44.0 | 52.7 | 67.5 | 36.7 | 36.7 | 48.3 |
| | GRPO | 68.3 | 56.9 | 62.1 | 65.0 | 44.0 | 52.0 | 68.3 | 40.7 | 36.7 | 49.1 |
| | ADPO | 71.3 | 59.3 | 64.8 | 68.3 | 48.0 | 52.0 | 69.2 | 39.3 | 39.5 | 50.8 |
| *Sample 8* | | | | | | | | | | | |
| Base | Major | 68.0 | 53.3 | 60.1 | 68.3 | 50.0 | 53.3 | 68.3 | 32.7 | 36.7 | 49.4 |
| | Base | 63.0 | 51.9 | 57.0 | 65.0 | 40.7 | 48.0 | 65.0 | 32.0 | 32.4 | 45.0 |
| | GRPO | 62.8 | 50.9 | 56.4 | 65.8 | 40.0 | 49.3 | 65.8 | 32.7 | 37.1 | 46.6 |
| | ADPO | 63.5 | 50.6 | 56.5 | 67.5 | 48.7 | 54.0 | 71.7 | 36.0 | 41.0 | 51.2 |
| GRPO | Major | 70.4 | 56.5 | 62.9 | 66.7 | 48.7 | 51.3 | 74.2 | 42.7 | 36.7 | 51.1 |
| | Base | 67.6 | 55.0 | 60.7 | 62.5 | 47.3 | 51.3 | 72.5 | 38.7 | 37.6 | 49.7 |
| | GRPO | 67.6 | 55.0 | 60.8 | 62.5 | 46.7 | 50.0 | 70.0 | 39.3 | 38.6 | 49.3 |
| | ADPO | 67.6 | 54.4 | 60.5 | 63.3 | 46.7 | 53.3 | 68.3 | 38.7 | 39.0 | 49.8 |
| ADPO | Major | 71.1 | 58.0 | 64.0 | 65.0 | 49.3 | 56.7 | 71.7 | 38.7 | 40.5 | 51.8 |
| | Base | 70.0 | 55.7 | 62.3 | 63.3 | 52.0 | 53.3 | 66.7 | 42.7 | 41.0 | 51.6 |
| | GRPO | 69.8 | 55.9 | 62.3 | 63.3 | 52.7 | 54.7 | 65.8 | 42.0 | 39.0 | 51.2 |
| | ADPO | 72.2 | 58.9 | 65.0 | 65.8 | 54.0 | 54.7 | 66.7 | 40.7 | 41.0 | 52.1 |
| *Sample 12* | | | | | | | | | | | |
| Base | Major | 67.4 | 55.0 | 60.7 | 69.2 | 52.0 | 50.7 | 70.8 | 38.0 | 36.7 | 50.7 |
| | Base | 63.7 | 51.1 | 56.9 | 59.2 | 47.3 | 51.3 | 64.2 | 30.0 | 33.8 | 45.8 |
| | GRPO | 62.4 | 51.1 | 56.3 | 58.3 | 45.3 | 49.3 | 63.3 | 30.0 | 35.2 | 45.2 |
| | ADPO | 62.6 | 48.5 | 55.0 | 65.0 | 52.7 | 53.3 | 70.0 | 40.0 | 35.2 | 50.6 |
| GRPO | Major | 70.7 | 57.2 | 63.4 | 64.2 | 50.0 | 51.3 | 73.3 | 43.3 | 39.5 | 51.7 |
| | Base | 70.0 | 56.3 | 62.6 | 63.3 | 48.0 | 54.0 | 68.3 | 42.7 | 41.0 | 51.2 |
| | GRPO | 69.3 | 56.7 | 62.5 | 63.3 | 48.7 | 52.7 | 67.5 | 42.0 | 40.5 | 50.8 |
| | ADPO | 69.6 | 55.2 | 61.8 | 64.2 | 48.0 | 52.7 | 69.2 | 43.3 | 41.0 | 51.3 |
| ADPO | Major | 72.0 | 58.3 | 64.6 | 65.8 | 50.0 | 53.3 | 75.0 | 36.7 | 39.0 | 51.2 |
| | Base | 70.7 | 56.5 | 63.0 | 62.5 | 52.0 | 54.7 | 70.0 | 41.3 | 41.4 | 52.0 |
| | GRPO | 71.3 | 56.9 | 63.5 | 63.3 | 53.3 | 52.7 | 70.8 | 41.3 | 43.3 | 52.6 |
| | ADPO | 71.7 | 59.8 | 65.3 | 67.5 | 53.3 | 54.0 | 71.7 | 38.7 | 40.5 | 52.3 |

Table 7: **Evaluation results on image grounding benchmarks.** Rows correspond to generators and columns correspond to verifiers. We use Qwen2-VL-7B as the base model, with GRPO and ADPO representing the finetuned models. Majority voting serves as the verifier baseline. Models are trained on RefCOCO training set and tested on ReasonSeg (out-of-domain).

| Generator | Verifier | Short query | | | Long query | | | Overall | | |
|---|---|---|---|---|---|---|---|---|---|---|
| | | gIoU | cIoU | ACC | gIoU | cIoU | ACC | gIoU | cIoU | ACC |
| *Sample 1* | | | | | | | | | | |
| Base | ✗ | 49.5 | 53.0 | 67.0 | 56.8 | 57.5 | 68.5 | 56.3 | 57.2 | 68.4 |
| GRPO | ✗ | 51.8 | 55.5 | 67.9 | 59.1 | 59.7 | 71.3 | 58.6 | 59.5 | 71.1 |
| ADPO | ✗ | 51.7 | 54.8 | **68.0** | 60.2 | 59.4 | **71.9** | 58.1 | 59.1 | **71.7** |
| *Sample 4* | | | | | | | | | | |
| Base | Major | 47.8 | 52.0 | 66.0 | 57.3 | 57.9 | **69.3** | 56.7 | 57.5 | **69.1** |
| | Base | 47.2 | 51.4 | 66.0 | 56.9 | 57.5 | 68.6 | 56.3 | 57.1 | 68.4 |
| | GRPO | 49.6 | 53.2 | 67.0 | 57.4 | 57.9 | 68.7 | 56.9 | 57.7 | 68.6 |
| | ADPO | 50.4 | 53.5 | **68.0** | 57.3 | 57.9 | 69.1 | 56.9 | 57.7 | **69.1** |
| GRPO | Major | 54.5 | 57.0 | 68.0 | 58.8 | 59.5 | 72.1 | 58.5 | 59.4 | 71.8 |
| | Base | 55.2 | 57.5 | 68.0 | 59.7 | 60.3 | 72.9 | 59.4 | 60.2 | 72.6 |
| | GRPO | 53.4 | 56.4 | **68.9** | 59.1 | 59.7 | 72.0 | 58.8 | 59.5 | 71.8 |
| | ADPO | 55.1 | 57.8 | 68.0 | 60.5 | 61.1 | **73.4** | 60.2 | 60.9 | **73.1** |
| ADPO | Major | 52.7 | **55.3** | 67.0 | 60.1 | 60.5 | 72.0 | 59.6 | 60.2 | 71.7 |
| | Base | 51.2 | 54.1 | 66.0 | 59.3 | 59.9 | 71.7 | 58.8 | 59.6 | 71.4 |
| | GRPO | 51.0 | 54.3 | 67.0 | 60.0 | 60.7 | 72.7 | 59.4 | 60.3 | 72.4 |
| | ADPO | 52.2 | 55.1 | 67.0 | 61.0 | 61.5 | **73.3** | 60.5 | 61.1 | **72.9** |
| *Sample 8* | | | | | | | | | | |
| Base | Major | 47.8 | 51.4 | **63.1** | 57.2 | 57.8 | 69.2 | 56.6 | 57.4 | 68.8 |
| | Base | 47.9 | 51.4 | 62.1 | 56.6 | 57.2 | 68.1 | 56.1 | 56.9 | 67.7 |
| | GRPO | 47.1 | 50.5 | 62.1 | 56.8 | 57.5 | 68.4 | 56.2 | 57.0 | 68.0 |
| | ADPO | 47.4 | 50.9 | 61.2 | 57.7 | 58.4 | **69.4** | 57.1 | 57.9 | **68.9** |
| GRPO | Major | 52.0 | 55.6 | **68.0** | 59.2 | 59.9 | 72.0 | 58.7 | 59.6 | 71.7 |
| | Base | 51.7 | 55.0 | 67.0 | 60.1 | 60.7 | 72.4 | 59.6 | 60.4 | 72.1 |
| | GRPO | 51.5 | 55.3 | **68.0** | 60.0 | 60.7 | 72.4 | 59.5 | 60.4 | 72.1 |
| | ADPO | 54.4 | 57.2 | 67.0 | 60.6 | 61.3 | **73.8** | 60.2 | 61.1 | **73.3** |
| ADPO | Major | 53.2 | 56.1 | 67.0 | 58.8 | 59.4 | 71.3 | 58.5 | 59.2 | 71.0 |
| | Base | 52.9 | 55.4 | 67.0 | 59.6 | 60.2 | 72.0 | 59.2 | 59.9 | 71.7 |
| | GRPO | 55.6 | 57.8 | **68.9** | 59.9 | 60.6 | 72.9 | 59.6 | 60.5 | 72.7 |
| | ADPO | 53.2 | 56.0 | 67.0 | 60.9 | 61.5 | **73.7** | 60.4 | 61.2 | **73.5** |
| *Sample 12* | | | | | | | | | | |
| Base | Major | 50.2 | 53.7 | 66.0 | 57.2 | 57.8 | **69.3** | 56.8 | 57.6 | **69.1** |
| | Base | 49.5 | 53.3 | **68.0** | 56.9 | 57.6 | 68.9 | 56.5 | 57.4 | 68.8 |
| | GRPO | 50.0 | 53.1 | 66.0 | 57.1 | 57.9 | 69.1 | 56.7 | 57.6 | 68.9 |
| | ADPO | 49.8 | 52.6 | 65.1 | 57.6 | 58.2 | 69.2 | 57.1 | 57.8 | 68.9 |
| GRPO | Major | 55.6 | 58.1 | **69.9** | 58.8 | 59.5 | 72.2 | 58.6 | 59.4 | 72.0 |
| | Base | 48.8 | 52.4 | 65.1 | 59.6 | 60.2 | 72.5 | 58.9 | 59.7 | 72.1 |
| | GRPO | 53.2 | 55.8 | 68.0 | 59.1 | 59.9 | 71.9 | 58.8 | 59.6 | 71.7 |
| | ADPO | 54.1 | 56.7 | 68.9 | 60.9 | 61.6 | **74.0** | 60.5 | 61.3 | **73.7** |
| ADPO | Major | 53.3 | 55.3 | 66.0 | 59.3 | 60.0 | 71.8 | 58.9 | 59.8 | 71.5 |
| | Base | 55.0 | 57.4 | **68.9** | 60.2 | 60.9 | 72.1 | 59.9 | 60.7 | 71.9 |
| | GRPO | 53.8 | 56.5 | 68.9 | 60.3 | 61.0 | 72.4 | 59.9 | 60.7 | 72.1 |
| | ADPO | 53.9 | 56.2 | 67.0 | 61.3 | 62.0 | **73.6** | 60.9 | 61.6 | **73.2** |

Table 8: **Evaluation results on mobile agent benchmarks.** Rows correspond to generators and columns correspond to verifiers. We use Qwen2-VL-7B as the base model, with GRPO and ADPO representing the finetuned models. Majority voting serves as the verifier baseline. Models are separately trained on AndroidControl and GUI Odyssey training sets and tested on respective test sets.

| Generator | Verifier | AndroidControl | | | GUI Odyssey | | |
| | | Type | Grounding | SR | Type | Grounding | SR |
| --- | --- | --- | --- | --- | --- | --- | --- |
| | | *Sample 1* | | | | | |
| Base | ✗ | 82.2 | 73.6 | 61.3 | 81.1 | 61.4 | 52.8 |
| GRPO | ✗ | **86.0** | **76.9** | **71.0** | 93.1 | **83.9** | **79.8** |
| ADPO | ✗ | 85.8 | 76.2 | 70.9 | **94.2** | 82.5 | 79.7 |
| | | *Sample 4* | | | | | |
| Base | Major | 76.3 | 68.1 | 56.0 | 76.9 | 55.3 | 46.5 |
| | Base | 72.1 | 67.8 | 52.5 | 75.3 | 55.6 | 45.2 |
| | GRPO | 74.9 | 71.4 | 57.7 | 75.3 | 55.2 | 45.3 |
| | ADPO | 76.4 | 74.5 | 60.7 | 75.1 | 55.7 | 45.6 |
| GRPO | Major | 85.5 | 77.2 | 71.0 | 94.7 | 83.9 | 81.3 |
| | Base | 85.4 | 77.2 | 71.0 | 94.3 | 83.8 | 81.0 |
| | GRPO | 85.4 | 77.3 | 70.8 | 94.4 | 83.7 | 80.7 |
| | ADPO | 85.6 | 77.7 | 71.2 | 94.5 | 84.0 | 81.4 |
| ADPO | Major | 86.6 | 77.1 | 71.6 | 93.9 | 83.5 | 79.8 |
| | Base | 86.4 | 76.4 | 71.0 | 94.7 | 84.2 | 81.2 |
| | GRPO | 86.4 | 77.9 | 72.0 | 94.7 | 84.2 | 81.1 |
| | ADPO | 86.3 | 79.5 | 72.7 | 94.7 | 84.5 | 81.6 |
| | | *Sample 8* | | | | | |
| Base | Major | 78.7 | 68.8 | 58.3 | 76.7 | 55.4 | 46.6 |
| | Base | 73.9 | 68.4 | 54.3 | 75.1 | 55.2 | 44.9 |
| | GRPO | 77.3 | 73.4 | 61.0 | 74.4 | 54.3 | 44.5 |
| | ADPO | 79.7 | 76.5 | 64.7 | 73.9 | 54.6 | 44.6 |
| GRPO | Major | 85.6 | 76.9 | 70.8 | 94.6 | 84.4 | 81.5 |
| | Base | 85.6 | 77.1 | 71.0 | 93.7 | 84.4 | 80.7 |
| | GRPO | 85.4 | 77.4 | 70.9 | 93.7 | 84.4 | 80.6 |
| | ADPO | 85.6 | 77.7 | 71.4 | 93.9 | 84.8 | 81.2 |
| ADPO | Major | 86.5 | 76.4 | 71.3 | 94.8 | 84.0 | 80.9 |
| | Base | 86.1 | 76.2 | 70.8 | 95.1 | 84.6 | 81.6 |
| | GRPO | 85.8 | 77.7 | 71.4 | 94.9 | 84.4 | 81.4 |
| | ADPO | **86.4** | **78.7** | **72.7** | **94.8** | **84.7** | **81.7** |
| | | *Sample 12* | | | | | |
| Base | Major | 78.9 | 68.7 | 58.3 | 76.9 | 55.5 | 46.9 |
| | Base | 73.4 | 67.9 | 53.6 | 74.5 | 55.5 | 44.6 |
| | GRPO | 76.8 | 73.2 | 60.7 | 73.5 | 54.3 | 44.0 |
| | ADPO | 79.2 | 76.7 | 64.5 | 72.9 | 53.8 | 43.6 |
| GRPO | Major | 85.6 | 77.4 | 71.1 | 94.5 | 84.0 | 81.1 |
| | Base | 85.6 | 78.0 | 71.4 | 93.0 | 84.0 | 79.9 |
| | GRPO | 85.4 | 77.5 | 70.9 | 93.1 | 83.9 | 79.7 |
| | ADPO | 85.7 | 77.9 | 71.5 | 93.2 | 84.2 | 80.3 |
| ADPO | Major | 86.6 | 76.7 | 71.9 | 94.4 | 83.7 | 80.5 |
| | Base | 86.5 | 76.3 | 71.6 | 94.8 | 84.6 | 81.5 |
| | GRPO | 85.4 | 78.6 | 71.9 | 94.6 | 84.1 | 81.1 |
| | ADPO | 86.3 | 78.9 | 72.9 | 94.4 | 84.5 | 81.4 |

Table 9: **Ablation of Preference reward.** Replacing the binary answer reward with our *preference reward* consistently strengthens self-verification (↑AUC/AP) and improves best of N selection performance on *Math*, *Grounding*, and *GUI Agent*.

| Domain | Method | Performance | | Verification | |
|---|---|---|---|---|---|
| | | pass@1 | best@8 | AUC | AP |
| Math | Binary reward | 61.9 | 63.4 | 0.509 | 0.640 |
| | Preference reward (ours) | **62.4** | **65.0** | **0.522** | **0.653** |
| Grounding | Binary reward | **71.8** | 72.6 | 0.636 | 0.789 |
| | Preference reward (ours) | 71.7 | **73.5** | **0.672** | **0.804** |
| GUI Agent | Binary reward | 70.6 | 70.6 | 0.609 | 0.765 |
| | Preference reward (ours) | **70.9** | **72.7** | **0.727** | **0.841** |

Table 10: **Ablation study on decoupled advantages.** Our proposed Dual Advantages method consistently outperforms reward aggregation in both task performance and solution verification across mathematical reasoning, grounding, and GUI agent tasks.

| Domain | Method | Performance | | Verification | |
|---|---|---|---|---|---|
| | | pass@1 | best@8 | AUC | AP |
| Math | Reward Aggregation | 61.5 | 63.1 | 50.4 | 61.8 |
| | Dual Advantanges (ours) | **62.4** | **65.0** | **52.2** | **65.3** |
| Grounding | Reward Aggregation | 70.4 | 71.0 | 0.641 | 0.774 |
| | Dual Advantanges (ours) | **71.7** | **73.5** | **0.672** | **0.804** |
| GUI Agent | Reward Aggregation | 71.0 | 70.4 | 0.542 | 0.715 |
| | Dual Advantanges (ours) | **70.9** | **72.7** | **0.727** | **0.841** |

