# OpenReview forum: "Calibrated Self-Verification for Multimodal LLMs via Advantage-Decoupled Preference Optimization"
_ICLR.cc/2026/Conference — ICLR 2026 Conference Withdrawn Submission_

### Official Review · Reviewer_sfa4 · 2025-10-23

**Soundness:** 3
**Presentation:** 3
**Contribution:** 2
**Rating:** 4
**Confidence:** 4

**Summary:**

This paper introduces ADPO (Advantage-Decoupled Preference Optimization), a reinforcement learning framework that jointly trains multimodal large language models (LLMs) for answer generation and self-verification. It replaces brittle binary rewards with preference-based ranking rewards that are more stable and informative under class imbalance. A decoupled optimization strategy separates the gradients for generation and verification to avoid interference and reward hacking. The approach improves verification calibration and enhances test-time scaling via parallel reasoning. Empirically, ADPO outperforms prior multimodal reasoning baselines across math, vision, and mobile-agent benchmarks, achieving notable percentage gains with modest extra training cost.

**Strengths:**

1. The paper effectively integrates generation and verification into a single policy using reinforcement learning.
2. The use of group-adaptive ranking and margin-based comparisons provides richer supervision than sparse binary signals, improving stability and calibration.
3. The decoupled advantage mechanism elegantly mitigates gradient interference and reward hacking—common issues in joint learning setups.

**Weaknesses:**

1. Why is this method applicable to MLLMs only? It is expected that the authors show some expts on text based LLMs too.
2. Results are shown on Qwen 7B parameter models. More variety in architecture and scale is expected. Also, applicability to open-ended multimodal generation or dialogue remains untested.
3. Very sure that a lot of space saving mechanisms have been used to reduce whitespace significantly around figures.
4. While quantitative improvements are shown, the paper lacks in-depth qualitative analysis or ablations explaining why specific components (e.g., group-adaptive thresholds) yield gains.
5. The reported improvements (e.g., +2–3%) are not very exciting relative to the added architectural and training complexity.

Typos:
1. intergrates on line 128.
2. Fig 2. "Perference"

**Questions:**

1. "Ablation studies show +0.03 average improvement in verification AUC/AP metrics." is this stat sig?
2. Eq. 10: Should answer and preference rewards be given equal weights? Fig 2 shows some lambdas though.
3. What is the latency implication of proposed method across tasks at test time?

---

### Official Review · Reviewer_9JTH · 2025-10-26

**Soundness:** 3
**Presentation:** 3
**Contribution:** 2
**Rating:** 4
**Confidence:** 4

**Summary:**

This paper introduces ADPO, an RL framework for MLLMs which aims to enhance MLLM reliability and test-time scalability through calibrated self-verification. It addresses limitations of existing test-time scaling methods and sparse binary rewards by training a unified policy for both answer generation and self-verification. Key contributions include a unified preference reward (providing stable, calibrated self-verification scores) and advantage-decoupled optimization (resolving gradient conflicts between generation and verification tasks). The method is validated on multimodal math, visual grounding, and mobile agent tasks, showing significant performance improvements.

**Strengths:**

1. ADPO proposes a unified generation and self-verification strategy, featuring an innovative unified preference reward and advantage-decoupled optimization.
2. Comprehensive evaluation and ablation studies across diverse multimodal tasks establishing the method's effectiveness and its components' contributions.
3. The paper is well-structured, written clearly, and features intuitive and informative figures, making it accessible to readers.

**Weaknesses:**

1. I believe Figure 1 displays limited information and fails to immediately highlight the key differences from other methods, particularly regarding ADPO's 'unified' characteristic. I suggest optimizing Figure 1 to better showcase the paper's core contributions to the readers.
2. While the preference reward addresses class imbalance, ADPO's reliance on within-batch contrastive learning might struggle if training batches consistently contain only correct or only incorrect samples. This homogeneity could diminish the preference reward's calibrating benefits for self-verification. More proactive data sampling or pre-processing to ensure a mix of positive and negative examples per batch could strengthen this aspect.
3. While the paper distinguishes itself from serial test-time scaling that involves "longer thinking tokens," it doesn't fully clarify the role or length variability of these generated reasoning traces within ADPO's framework, especially given the "no-think" results cited in related work. Further analysis on how ADPO's self-verification interacts with, or is influenced by, these generated thinking processes would be valuable.

**Questions:**

1. Was initializing the model with some basic self-verification ability (e.g., via supervised learning) considered? This could potentially aid convergence or score calibration. If not, what was the rationale, and were any related training challenges observed?
2. ADPO's evaluation uses specific Qwen-VL models. Since the initial scoring ability relies on the base model, do the authors expect ADPO's effectiveness to generalize well to fundamentally different MLLM architectures or significantly larger models? What insights can be offered regarding ADPO's scalability and robustness across diverse underlying MLLM capabilities?
3. The paper could better quantify and discuss the significant N-fold computational overhead at inference time compared to a single-pass generation, and how this trade-off impacts practical deployment, especially for latency-sensitive applications.
4. How does the model handle cases that self-verification scores exhibit exceptionally high consistency, yet high-scoring outputs do not represent the optimal output, or when the correlation between scores and output quality is low? The paper could include several case studies to better illustrate the advantages of a unified self-verity approach.

---

### Official Review · Reviewer_qXog · 2025-10-28

**Soundness:** 2
**Presentation:** 2
**Contribution:** 2
**Rating:** 4
**Confidence:** 4

**Summary:**

The authors propose an RL framework, termed ADPO, for training MLLMs to generate answers and self-verify via preference reward and decoupled advantages. Superior experimental results are achieved on multimodal math reasoning, image grounding and mobile agent tasks.

**Strengths:**

(1)The test time scaling of MLLMs is crucial for enhancing the reliability and robustness of MLLMs. And the idea of using the self-verification task to improve the effectiveness of TTS training is novel and well-motivated.

(2)For the issue of gradient interference, a dual-advantage optimization method is proposed within the GRPO framework.

(3)The dense, contrastive preference reward is insightful to overcome the limitations of binary reward in model training.

**Weaknesses:**

Cons:

(1) The proposed Generation-Verification framework is alike the well-known actor-critic RL method, where the verifier serves as the the critic for scoring the generated answer. Meanwhile the verifier is also the reward model in well-known RLHF paradigm. Could you provide more discussions to clarify the difference between the proposed method with the other two learning paradigms?

(2) The Introduction section describes ‘s+’ and ‘s-’ as two group-adaptive thresholds, but this terminology and symbolism are not consistent in the subsequent technical sections. Although it seems that the calculation method for ‘s-’ is represented by ‘u_i’ in Eq. (7) (as the Eq. (8) and Eq.(9) seem to find the negative sample pairs, is my understanding correct?), the method for calculating ‘s+’ and how it is applied to compute rewards for negative responses are not clear?

(3) Figure 1 is somewhat unclear, what’s the meaning of the numbers in the pink areas? How to represent the concept of token masks?

(4) Maybe the unified objective in Eq. (10) should have a hyperparameter to control the trade-off between hard binary reward and preference reward?

(5) In Eq. (11), what is the shape of the token mask M^a and M^p? Due to the separative characteristic, can we simply represent M^a as 1-M^p?

(6) In ablation studies, compared to Math and Grounding tasks, the effectiveness of the proposed Preference Reward and Decoupled Advantages methods in the mobile agent evaluation benchmark is highly significant. What are the reasons for this?

(7) In the experiments, some common MLLM evaluation benchmarks, such as MME, MMbench, and MME-RealWorld, were not used to evaluate the proposed method. It is suggested that these benchmarks be included for comparison to facilitate a contrast with other post-training methods for MLLMs, such as MM-RLHF.

(8) The authors should adjust the spacing between the captions of all figures and the main text. For instance, the caption for Figure 1 appears too close to the text, and increasing the spacing would improve readability and overall presentation.

**Questions:**

Please see the Weaknesses.

---

### Official Review · Reviewer_fBgY · 2025-10-30

**Soundness:** 2
**Presentation:** 2
**Contribution:** 2
**Rating:** 2
**Confidence:** 4

**Summary:**

This paper proposes ADPO, a  reinforment-learning framework that trains a multimodal LLM to generate both answer to the question as well as a score to self-verify the geneation quality, then calculate the answer and verification advantages separately. The authors conduct experiments on math, grounding and GUI benchmarks to evaluate the proposed method.

**Strengths:**

- The proposed method provide a solution for best-of-N sampling without external reward feedback.
- To the reviewer's knowledge, the design of preference reward is novel. Ablation studies is provided to validate the effectiveness of the design.

**Weaknesses:**

- Each model is seperatedly trained on a single task. The method’s generalization across *multiple* tasks remains unvalidated.
- The evaluation scope is limited — for example the authors did not include relevant benchmarks such as MathVerSe, MM‑Star, MMVet or AI2D.
- The performance gains are modest. For instance, in Table 4, the best-of-1 sampling is worse than the vanilla baseline GRPO, and the improvements at sample 4 and sample 8 are only ~0.2 and ~0.5 points respectively.

**Questions:**

- When doing best-of-N sampling, how do you apply GRPO-trained models as verifiers?
- How does self-generated score compare to entropy-based confidence score? Could the authors provide a small ablation study to validate this comparison?

---

### Note · Authors · 2025-11-12

I have read and agree with the venue's withdrawal policy on behalf of myself and my co-authors.